# Rationally Designed Ternary Deep Eutectic Solvent Enabling Higher Performance for Non-Aqueous Redox Flow Batteries

**Ping Lu [1,2], Peizhuo Sun [1,2], Qiang Ma [1], Huaneng Su [1], Puiki Leung [3], Weiwei Yang [4,\*] and Qian Xu [1,\*]** 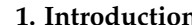

1. Institute for Energy Research, Jiangsu University, Zhenjiang 212013, China; flightinglp@163.com (P.L.); 2221906026@stmail.ujs.edu.cn (P.S.); maqiang@ujs.edu.cn (Q.M.); suhuaneng@ujs.edu.cn (H.S.)
2. School of Energy and Power Engineering, Jiangsu University, Zhenjiang 212013, China
3. MOE Key Laboratory of Low-Grade Energy Utilization Technologies and Systems, Chongqing University, Chongqing 400030, China; p.k.leung@soton.ac.uk
4. School of Energy and Power Engineering, Xi'an Jiaotong University, Xi'an 710049, China
* Correspondence: yangww@mail.xjtu.edu.cn (W.Y.); xuqian@ujs.edu.cn (Q.X.)

**Abstract:** Redox flow batteries hold promise as large-scale energy storage systems for off-grid electrification. The electrolyte is one of the key components of redox batteries. Inspired by the mechanism involved in solvents for extraction, a ternary deep eutectic solvent (DES) is demonstrated, in which glycerol is introduced into the original binary ethaline DES. Redox pairs (active substance) dissolved in the solvent have low charge transfer resistance. The results show that the viscosity of the solvent with the ratio of choline chloride to ethylene glycol to glycerol of 1:2:0.5 decreases from 51.2 mPa·s to 40.3 mPa·s after adding the redox pair, implying that the mass transfer resistance of redox pairs in this solvent is reduced. Subsequent cyclic voltammetry and impedance tests show that the electrochemical performance with the ternary DES as the electrolyte in redox flow batteries is improved. When the ratio of 1:2:0.5 ternary DES is used as the electrolyte, the power density of the battery (9.01 mW·cm$^{-2}$) is 38.2% higher than that of the binary one (6.52 mW·cm$^{-2}$). Fourier transform infrared spectroscopy further indicates that the introduction of glycerol breaks the hydrogen bond network of the solvent environment where the redox pair is located, unraveling the hydrogen bond supramolecular complex. Rational solvent design is an effective strategy to enhance the electrochemical performance of redox batteries.

**Keywords:** deep eutectic solvent (DES); hydrogen bond; ternary solvent; redox flow battery; battery performance

## 1. Introduction

In the face of fossil energy shortage and environmental pollution, it is imperative to exploit renewable energies. However, the problem of discontinuous and unstable power output is still associated with renewable energy, the solution or mitigation of which requires low-cost, high energy density, and safe energy storage systems [1]. The performance of some battery energy storage technologies is shown in Table 1 [2–5]. Among them, the redox flow battery, which stores energy in the form of a liquid phase externally, possesses an independent design of energy and power density, low pollution emission in large-scale energy storage. It is considered to be the one of most emerging technologies in grid energy storage technology [6,7]. For the past few years, the redox flow battery has made continuous progress in critical materials such as ion exchange membranes [8], electrodes, and redox pairs, among which the all-vanadium redox flow battery proposed by Skyllas-Kazacos et al. [9] has entered the stage of commercial application [10]. However, the price of vanadium ions is relatively high. Furthermore, the strict requirements of temperature and concentration of vanadium ions restrict the energy density of the electrolyte [11,12].

**Table 1.** The performance of some battery energy storage technologies.

| Technology | | Power Density (Wkg$^{-1}$/kWm$^{-3}$) | Energy Density (Wkg$^{-1}$/kWhm$^{-3}$) | Lifetime (Cycle) | Drawbacks |
|---|---|---|---|---|---|
| Physical energy storage | | /0.1–5000 | /0.2–80 | >5000 | site specific, high cost |
| Supercapacitors | | 0.1–100/ 40,000–120,000 | 0.1–15/10–12 | $5 \times 10^5$ | cryogenics, high cost |
| Lithium-ion batteries | | /1300–10,000 | /250–625 | $4 \times 10^3$ | safety, high cost |
| Redox flow batteries | Zn-Br$_2$ | 50–150/1–25 | 60–80/20–35 | >2000 | low energy density |
| | Vanadium | /0.5–2 | /20–35 | $13 \times 10^3$ | |

An electrolyte, which is made by dissolving redox pairs in a solvent, is one of the key components of redox batteries. Although considerable research has been devoted to developing and studying redox pairs, little attention is paid to the designability of solvents. Previously, water-based solvent electrolytes have been used in redox flow batteries owing to their strength of high ionic conductivity and non-flammability [13]. The solvent has been innovated from the aqueous system to the non-aqueous system lately. Non-aqueous solvents can provide a wider potential window [14] and the possibility of dissolving more redox pairs. In non-aqueous systems, the toxicity and flammability of organic solvents [15] and the disadvantages of expensive ionic liquids and complex synthesis [16] limit the large-scale application of these two solvents in redox flow batteries. Furthermore, the design principles of solvents should conform to the current development trend of green chemistry [17].

Abbot et al. presented the concept of "deep eutectic solvent" (DES) in 2004 [18]. DES is a eutectic mixture composed of a hydrogen bond acceptor (HBA, such as quaternary ammonium salts) and a hydrogen bond donor (HBD, such as carboxylic acids and polyols) in a certain stoichiometric ratio. DES is a new type of non-aqueous solvent with solubility [19]. It has been observed that DES has almost the same advantages as ionic liquids. In addition, DES has the characteristic of biodegradability and synthesis without purification to save costs. In this sense, it is a promising "green" solvent. DES has been used in many fields, such as catalyst precursors [20,21] and solvents for extraction and separation [22,23], materials chemistry [24,25], pretreatment and separation solvent of cellulose in biomass [26,27], electrochemistry [28–30], etc. However, most research is still concentrated upon binary deep eutectic solvents.

Traditional hydrophilic polyol-based DES such as ethylene glycol and choline chloride; glycerol and choline chloride, etc., have relatively high conductivity. The solubility of some DES to common metal oxides is even comparable to that of acids. Nevertheless, the relatively high viscosity of DES is an inescapable problem. Most of the research mentioned deals with adjusting the physical properties of DES by changing hydrogen bond donors or adding additives. Recently, there have been a few studies on ternary deep eutectic solvents (TDES). In 2018, based on the acidic multisite coordination theory, Xia et al. designed a TDES by introducing AlCl$_3$·6H$_2$O as an anion donor and active acidic site holder into the deep eutectic solvent synthesized by choline chloride and glycerol. The fractionation efficiency of lignin was significantly increased from 3.61% to 95.46% [31]. After that, in 2020, based on the TDES prepared by the Xia team, Ji et al. combined it with ultrasonic and microwave to further optimize the best ultrasonic frequency, molar ratio, metal chloride, and heating method, so that the TDES delignification rate was as high as 90% [32]. It can be learned that AlCl$_3$·6H$_2$O (redox pairs in redox flow batteries), as a solvent component, is involved in the synthesis of DES, that is, it is involved in the construction of a hydrogen bond network in the solvent.

DES has also been applied in redox flow batteries. Yu et al. mixed urea and AlCl$_3$ at a molar ratio of 1:1.3 to prepare an Al-DES electrolyte. Subsequently 1, 2-dichloroethane was added to reduce viscosity and improve conductivity [33]. After that, using the same

Al-DES electrolyte as the anolyte and $FeCl_3 \cdot 6H_2O$/urea/ethylene glycol as the catholyte, it is concluded that the iron complex can be dissociated when ethylene glycol is used as an additive [34]. It was found that $FeCl_3 \cdot 6H_2O$, which also acts as redox pairs, is not a component of the DESs in the flow battery, but a substance through the oxidation–reduction reaction to achieve energy storage in this work.

Since redox pairs participate in the construction of solvent hydrogen bond networks in the field of extraction, it is impossible that they will not participate in the hydrogen bond network because of the change of the research field, which is the nature of redox pairs. The redox reaction of redox pairs (inorganic salt hydrate) in solvent depends on the metal center and coordination environment. However, the existing form of the redox pairs in a non-aqueous environment is rarely studied.

The addition of redox pairs changes the solvent environment. In order to weaken the solvent environment and redox pairs through the hydrogen bond network formation of macromolecular structure caused by the increase of migration resistance. In this work, glycerol was introduced because of its low material cost, low viscosity as a low-chain alcohol, and has no effect on the pH to form a more stable TDES. TDES, as a solvent that can be designed [35] by adjusting the molar ratio of choline chloride, ethylene glycol, and glycerol, has achieved the goal of microscopically affecting the electronegativity between HBD and HBA and the bond length of the hydrogen bond and other parameters. At the same time, major aspects such as the conductivity, viscosity, stability, and temperature adaptability of the electrolyte were studied. Redox pairs can have a positive impact on electrochemical performance in the presence of crystal water [36]. We used crystalline hydrate as redox pairs and assembled an iron–vanadium flow battery. The transport performance and electrochemical performance of the binary eutectic solvent system and alcohol-based ternary DES systems were tested. Furthermore, theory and the Fourier transform infrared spectroscopy indicated that the introduction of glycerol breaks and reconstructs the hydrogen bond network of the solvent environment of the redox pair.

## 2. Results

### 2.1. Physical and Electrochemical Properties of TDES

The cyclic voltammetry curves of pure TDES at different temperatures were recorded. The electrode potential ranged from $-2.0$ V to 0.85 V. It can be seen from Figure 1a that compared with the traditional water-based flow battery, the blank TDES as an electrolyte has a wider electrochemical window and no redox peak in the scanning region at room temperature. When the temperature rises from 45 °C to 55 °C, the current increases rapidly at 0.7 V. TDES decomposed significantly from a slightly fluctuating state, which proves that the solvent has a wider and more stable potential window than water below 45 °C. The decomposition potential has no influence on the subsequent study of redox pairs. At the same time, the asymmetric ion on the charge delocalization of DES reduces the lattice energy and freezing point of DES [37], which also proves that the solvent has more resistance to low temperature than water.

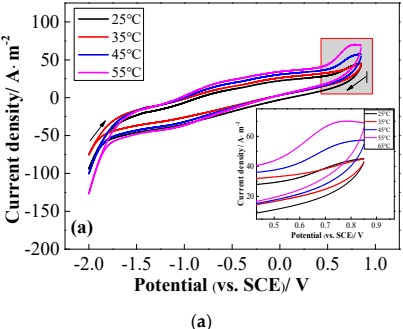

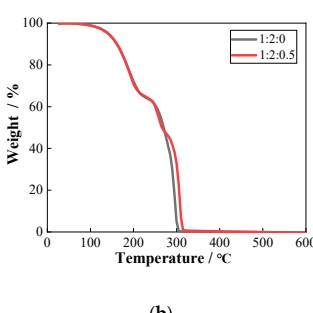

(a)     (b)

**Figure 1.** (**a**) Cyclic voltammograms of 1:2:0.5 TDES at different temperatures with a scan rate of 100 mV·s$^{-1}$ at room temperature (25 °C); (**b**) TG curves of BDES and 1:2:0.5 TDES.

The effect of glycerol addition on the thermal properties of electrolytes was evaluated by thermogravimetric analysis (TG) as shown in Figure 1b. The thermogravimetric curves of BDES(TDES) show two mass drops, which means the decomposition of 2 organic components. The thermogravimetric curves of TDES show three mass drops, which means the decomposition of 3 organic components. The hydrogen bond network of the two solvents begins to break slightly at 4 °C and appears to obviously fracture above 10 °C. The decomposition rate of the hydrogen bond is accelerated above 150 °C. The components of the solvents began to decompose as the temperature continued to rise. Both solvents have better thermal stability.

Viscosity has a great influence on the mass transfer in DESs, which is an important parameter that determines the efficiency of flow batteries. Viscosity is closely related to the intermolecular interaction of DES components. The stronger the hydrogen bonds between molecules are, the bigger the viscosity of the prepared DES is [38]. The viscosity of DES is easily affected by the size of the cationic, anionic components, and the voids in the liquid [39]. Through the design of controlling the molar ratio of glycerin, the results suggest that the viscosity of the solvent in which the ratio of choline chloride to ethylene glycol to glycerol of 1:2:0.5 decreases from 51.2 mPa·s for the binary eutectic solvent to 40.3 mPa·s after adding the redox pair (FeCl$_3$·6H$_2$O) in Figure 2. The viscosity did not change much without the addition of the redox pair. The original hydrogen bond network structure was appropriately destroyed by the introduction of glycerin and the low viscosity is obtained, which means that this design establishes a good balance and is beneficial to the flow battery. On the basis of controlling the viscosity of DES, the dynamics and efficiency are simultaneously improved. In the binary system, when the FeCl$_3$·6H$_2$O redox pair is added, the hydrogen bonds of anions and cations in FeCl$_3$·6H$_2$O can attract the -OH on the ethylene glycol to compete with choline chloride, forming a large number of relatively strong anion or neutral hydrogen bond, simultaneously supplying multiple sites of Cl. The coordination between CI and ethylene glycol occupies free sites, forming hydrogen bonds and van der Waals bonds of different strengths. The hexacoordinate FeCl$_3$·6H$_2$O has an electron-withdrawing group, which can interact with ethylene glycol to form a single iron ligand. The ethylene glycol acts as a "bridge" to connect -OH, choline chloride with FeCl$_3$·6H$_2$O to form a chloride ion–iron metal ion–ethylene glycol supramolecular complex [31]. When the battery is running, the large and complex molecular structure plays a great role in mass transfer resistance, and its viscosity is relatively large in Figure 2. After introducing the glycerol with different structures, the original hydrogen bond network structure is destroyed by microscopically adjusting the electronegativity difference between the hydrogen bond donor, the hydrogen bond acceptor, and the length of the hydrogen bond. Then, the energy of TDES was minimized by ChemBio 3D Ultra and a 3D structure diagram of Figure 3 was drawn. After the redox pair is added at a molar ratio of 1:2:0.5, it is speculated that it may not greatly destroy the hydrogen bond network system of the TDES. The redox pair is in a free state rather than a supramolecular complex, which shows good transport performance in the flow battery. The conductivity of DES in the figure did not change much in comparison with viscosity.

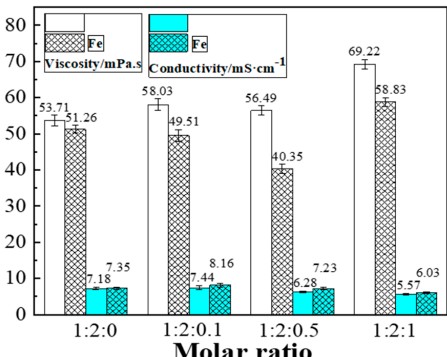

**Figure 2.** Conductivity and viscosity of the DESs at 25 °C without and with FeCl$_3$·6H$_2$O.

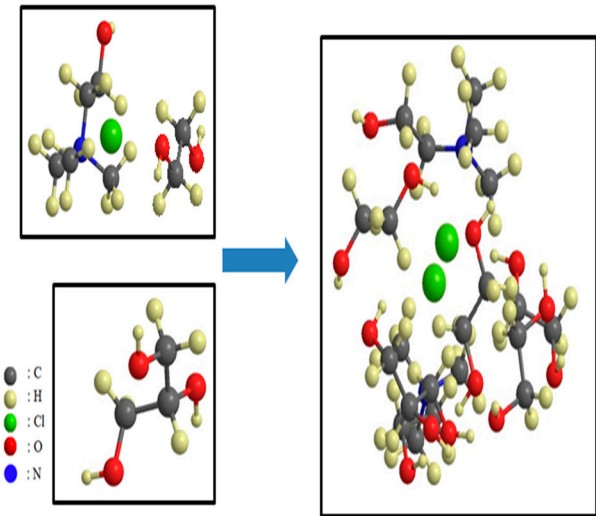

**Figure 3.** A 3D structure diagram of the 1:2:0.5 TDES.

### 2.2. Effect of TDES on Fe (III)/Fe (II)

At two different scanning rates in Figure 4, with different additions of glycerol, the peak potential difference decreases at 1:2:0.5, the reversibility is the strongest compared with other ratios, and the peak current density is the largest. With the sweep rate from 50 mV·s$^{-1}$ to 100 mV·s$^{-1}$, the peak current densities of 1:2:0.1 and 1:2:0.5 increase from 55.00 A·m$^{-2}$, 59.59 A·m$^{-2}$ to 55.59 A·m$^{-2}$, 79.46 A·m$^{-2}$, respectively. The increase rate of 1:2:0.1 is obviously weaker than that of 1:2:0.5. It is speculated that the addition of a small amount of glycerol has no effect on the reconstruction of the hydrogen bond network after the sweep rate increases. It reveals the importance of the appropriate molar ratio of the components for the construction of a hydrogen bond network. In the above experiment, the background current of electrolyte lower than 0.1 mol·L$^{-1}$ is larger. In order to avoid background subtraction and simplify data analysis, an electrolyte with a concentration of 0.1 mol·L$^{-1}$ was selected. As can be seen from Figure 5, the background current was only about 2% of the redox peak, which did not affect the analysis of battery performance. The data in Table 2 are the iron ion diffusion coefficient calculated according to the Randles–Sevcik equation. Detailed calculations are shown in Figure S1 of the Supplementary. With the addition of glycerol, the iron ion diffusion coefficient reaches the maximum value of 8.29 × 10$^{-7}$ at the molar ratio of 1:2:0.5 and improves by about 1.5 times.

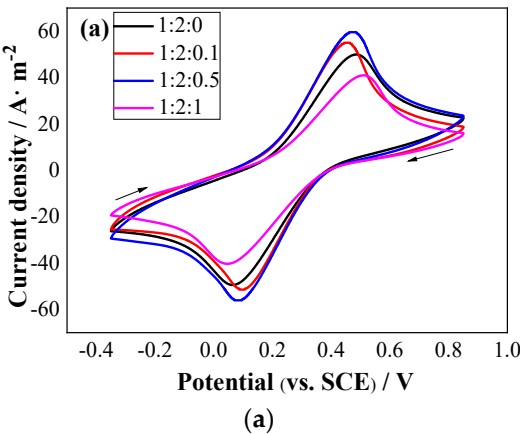

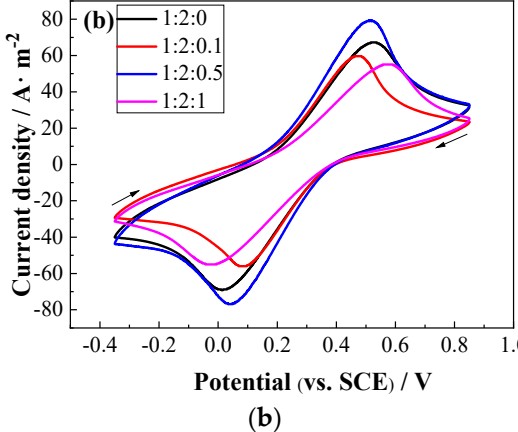

**Figure 4.** Cyclic voltammogram of the DESs with 0.1 mol·L$^{-1}$ FeCl$_3$·6H$_2$O at the scan rates of (**a**) 50 mV·s$^{-1}$; (**b**) 100 mV·s$^{-1}$.

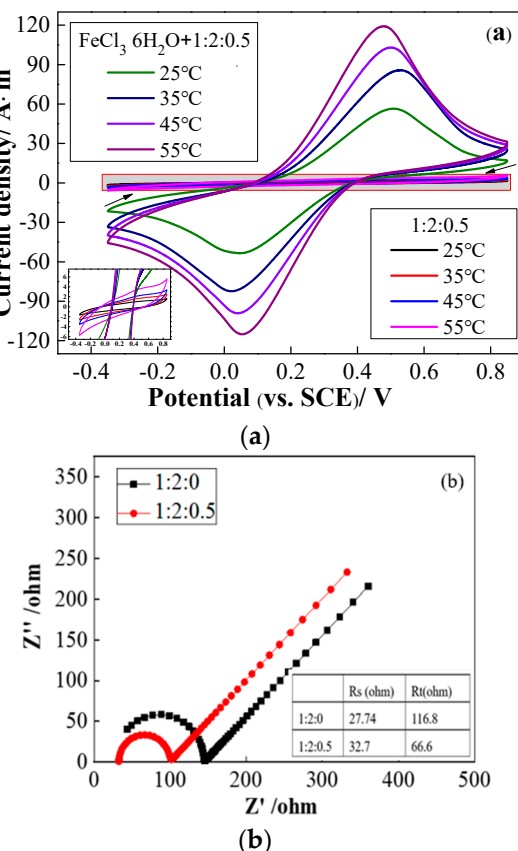

**(a)**

**(b)**

**Figure 5.** (**a**) Cyclic voltammogram of the DESs with 0.1 mol·L$^{-1}$ FeCl$_3$·6H$_2$O in different temperatures at the scan rates of 50 mV·s$^{-1}$; (**b**) electrochemical impedance spectroscopy of the DESs with 0.1 mol·L$^{-1}$ FeCl$_3$·6H$_2$O dissolved.

**Table 2.** Diffusion coefficient in different molar ratio of the DESs with 0.1 mol·L$^{-1}$ FeCl$_3$·6H$_2$O.

| $n_{choline\ chloride}$:$n_{ethylene\ glycol}$:$n_{glycerol}$ | $D_{re}$/cm$^2$·s$^{-1}$ |
| --- | --- |
| 1:2:0 | $5.54 \times 10^{-7}$ |
| 1:2:0.1 | $7.36 \times 10^{-7}$ |
| 1:2:0.5 | $8.29 \times 10^{-7}$ |
| 1:2:1 | $3.90 \times 10^{-7}$ |

For optimizing the ratio of the ternary system electrolyte, its electrochemical performance at different temperatures was investigated. From the CV curve in Figure 5a, it can be clearly observed that different temperatures can significantly affect the electrochemical properties of iron ions in TDES. The anode current i$_{pa}$ is 56.51 A·m$^{-2}$ at room temperature, and the peak current value reaches 119.22 A·m$^{-2}$ when the temperature rises to 55 °C. As the temperature increases, lower viscosity and higher ionic conductivity can be obtained. The peak potential difference (△E) decreases as the temperature increases, which means the reversibility of the reaction increases. According to the analysis of the Arrhenius equation, it can be seen that the rise of temperature intensifies the movement of ions in the solution, resulting in effective high-frequency collisions between ions, which also means that the ions are removed from the solvent film, escaping from the lattice and further accelerating the exchange current between the ions and electrodes in the electrolyte. Ions are more likely to overcome the activation energy [40], making electrochemical reactions more reversible.

The impedance spectrum shown in Figure 5b is composed of a semicircle in the high-frequency region and a straight line in the low-frequency region [41], which indicates that the redox reaction of iron ions is controlled by a mixture of electrochemical reactions and

diffusion steps. The semicircle part reflects the $R_s$ (ohmic resistance) and $R_t$ (charge transfer resistance) in the transfer reaction between the electrode and the electrolyte interface. It can be seen that the resistance of the electrolyte and electrode in the ternary solution is relatively larger than that in the binary solution while the charge transfer resistance is greatly reduced. It means that the charge transfer process of the electrolyte in the solution is accelerated, and the electrochemical reaction rate is increased. Meanwhile, the reconstruction of the hydrogen bond network and the dissociation of supramolecular complexes mentioned above are confirmed. This conclusion is consistent with the results of the CV diagram, which can be attributed to the reconstruction of the hydrogen bond network of the solvent.

### 2.3. Influence of TDES on Fe–V Redox Flow Battery

In order to verify the influence of BDES and TDES as electrolytes on battery performance, 0.1 mol·L$^{-1}$, the same positive and negative redox pairs, namely FeCl$_2$·4H$_2$O and VCl$_3$, were prepared in two different solvent systems, respectively. Separate the positive and negative chambers with Nafion212 membrane, use carbon felt as the electrode, and assemble the iron–vanadium flow battery as shown in Figure 6a. The experiment is set up from the open-circuit voltage of 1.02 V, and the discharge current gradually increases until the discharge voltage drops to zero. The polarization curve obtained by the test is shown in Figure 6b. It can be observed that when the ternary deep eutectic solvent of 1:2:0.5 is used as the solvent, the power density of the battery system (9.01 mW·cm$^{-2}$) is higher than that of the binary (6.52 mW·cm$^{-2}$). It can be inferred that TDES with a suitable molar ratio has better solubility than BDES containing supramolecular complexes for redox pairs.

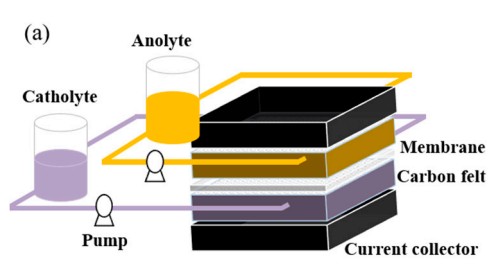
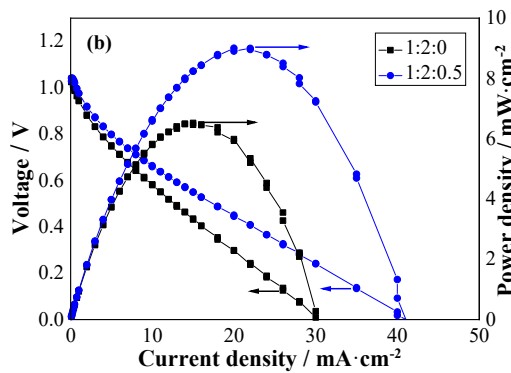

**Figure 6.** (**a**)Assembly of ferrovanadium flow battery; (**b**) power density and I–V curves of ferrovanadium flow battery with the two different ratios of DESs.

The charge–discharge current density of the battery was set to 2 mA·cm$^{-2}$, and the charge–discharge time was set to 30 min. According to the following Equations (1)–(3), since the charge and discharge currents are both 2 mA·cm$^{-2}$, the default Coulomb efficiency (CE) is about 100%, and the voltage efficiency (VE) is approximately equal to the energy efficiency (EE). It can be seen from Figure 7a that the charging voltage decreased significantly, the discharging voltage increased, and the energy efficiency of the battery increased from 63.06% to 77.41% after glycerin was added in the first charge–discharge cycle. In the fifth cycle, the energy efficiency dropped to 65.12% when glycerol was added, but it was still relatively stable, while the energy efficiency without glycerol began to drop sharply in Figure 7b. The results show that the addition of glycerin to BDES can enhance the redox to electrochemical generation in the graphite felt electrode of the battery. Thus, the energy efficiency of the battery is improved and the charge–discharge performance of the battery is improved.

$$CE = Q_{dis}/Q_{ch} \times 100\% = It_{dis}/It_{ch} \times 100\% = t_{dis}/t_{ch} \times 100\% \tag{1}$$

$$VE = V_{dis}/V_{ch} \times 100\% \tag{2}$$

$$EE = CE \times VE \tag{3}$$

where $Q_{dis}$ and $Q_{ch}$ represent the discharge capacity and the charge capacity, respectively, $t_{dis}$ is the discharge time, and $t_{ch}$ is the charge time, $V_{dis}$ and $V_{ch}$ stand for the voltage of discharge and charge and "$I$" is current.

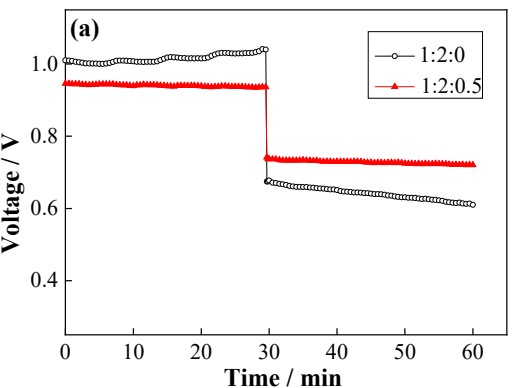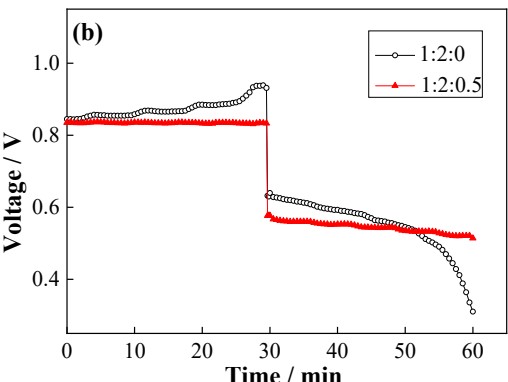

**Figure 7.** (**a**)The second and (**b**) fifth charging–discharging curves of ferrovanadium flow battery in two different ratios of DESs.

### 2.4. FT-IR Spectroscopy

According to the analysis of Figure 8, near 3300 cm$^{-1}$, the stretching frequency of the hydroxyl group in the ternary pure solvent is lower than that in the binary pure solvent, which means that the force of the intermolecular and intramolecular hydrogen bonds is weakened [42]. With the addition of FeCl$_3$ 6H$_2$O redox pairs, a peak of Fe-OH appears near 1698 cm$^{-1}$ [31]. Specifically, the peak intensity Fe-OH of the binary electrolyte is higher than that of the ternary electrolyte. It can be inferred that the chloride ion–metal cation–ethylene glycol supramolecular complex is formed in the binary system, which impedes the migration of metal cation. The introduction of glycerol destroys the original iron-based hydrogen bond network structure, breaks the complex supramolecular complex, and releases some small metal ions, which reduces the migration resistance of ions in the redox flow battery. The results are consistent with those of impedance analysis.

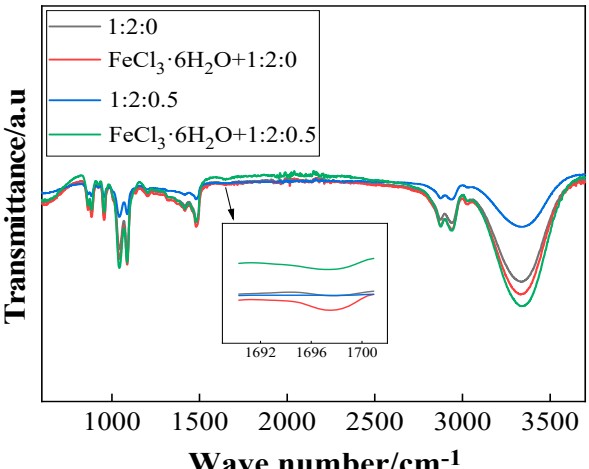

**Figure 8.** FT-IR spectra of the two different ratios of DESs without and with 0.1 mol·L$^{-1}$ FeCl$_3$·6H$_2$O dissolved.

## 3. Materials and Methods

### 3.1. Preparation of Electrolyte

The DES, commonly known as "ethaline" was fabricated by blending choline chloride (Aladdin, 98%) with ethylene glycol (Sinopharm Reagent 107-21-1, 99%) at the molar ratio of 1:2. Glycerin (Sinopharm Reagent 56-81-5, 99%) was added and stirred evenly into the DES electrolyte according to 1:2:0 (1:2:0 is defined as binary DES (BDES) in the following sections), 1:2:0.1, 1:2:0.5, 1:2:1 (TDES), and no immiscibility phenomena were observed after 1 h. The active material $FeCl_3 \cdot 6H_2O$ was added to the ethaline at a concentration of 0.1 mol·L$^{-1}$, which was heated and stirred at 100 °C to make the electrolyte. The electrochemical and physical properties were measured at different molar ratios.

### 3.2. Electrochemical Measurement

The viscosity and conductivity of the electrolyte have a great influence on the transmission of active materials inside the battery, so they are important physical parameters of the battery. In this experiment, a DV-2 + PRO digital viscometer (Shanghai Nirun Intelligent Technology Co., Ltd., Shanghai, China) and a DDS-307A digital conductivity meter (Shanghai Yidian Instrument Co., Ltd., Shanghai, China) were used for testing.

Cyclic voltammetry (CV) and electrochemical impedance spectroscopy (EIS) were determined using a Chenhua CHI600 electrochemical workstation. Deep eutectic solutions are prepared in different proportions as described in the previous section "Electrolyte preparation". The electrochemical properties of the electrolyte were studied with a three-electrode structure comprised of a 5 mm diameter glassy carbon working electrode, a platinum counter electrode, and a saturated calomel reference electrode (SCE) filled with a salt bridge of saturated potassium chloride solution. Before the experiment started, the glassy carbon electrode was polished with 50 nm aluminum powder and then rinsed with deionized water. Prior to the experimental measurement, nitrogen gas was introduced into the electrolyte for about 10 min to remove the oxygen dissolved in the solution.

During the electrochemical impedance spectroscopy measurement, the frequency range changes from 0.01 Hz to 100,000 Hz.

## 4. Conclusions

In this paper, it is found that redox pairs play an important role as free ions that need to migrate and take part in redox reactions, but they actually participate in hydrogen bond recombination and are bound in the synthesis of deep eutectic solvents (DES). Glycerol is introduced to disturb the formation of hydrogen-bonded supramolecular complexes by iron ions and binary eutectic solvents. Then the bonded iron ions are released to migrate freely and undergo electrochemical reactions, thus promoting the performance of the flow battery. The main findings are shown below:

(1) The addition of glycerin has a positive effect on physical properties (viscosity and conductivity, etc.).
(2) With 1:2:0.5-ternary deep eutectic solvents, the power density of the battery (9.01 mW·cm$^{-2}$) is higher than that of the binary system (6.52 mW·cm$^{-2}$). When glycerin is added, charging voltage decreases significantly, the discharging voltage increases, and the energy efficiency of the battery increases from 63.06% to 77.41%.
(3) Combining the experimental results and FT-IR spectra, it can be inferred that the electron migration of metal cation is hindered by the formation of the chloride ion–metal cation–ethylene glycol supramolecular complex in the binary system.

**Supplementary Materials:** The following supporting information can be downloaded at: https://www.mdpi.com/article/10.3390/pr10040649/s1, Figure S1: The curve of peak current density and the square root of scanning rate.

**Author Contributions:** Conceptualization, P.L. (Ping Lu) and Q.X.; Funding acquisition, W.Y. and Q.X.; Investigation, P.L. (Ping Lu), Q.M. and H.S.; Methodology, P.L. (Ping Lu) and P.S.; Validation,

P.L. (Ping Lu); Writing—Original draft, P.L. (Ping Lu); Writing—Review and editing, P.L. (Ping Lu), P.L. (Puiki Leung), W.Y. and Q.X. All authors have read and agreed to the published version of the manuscript.

**Funding:** The work described in this paper was fully supported by Grants from the NSFC, China (No. 51676092), State Key Laboratory of Engine at Tianjin University (No. K2020-14), Six-Talent-Peaks Project in Jiangsu Province (2016-XNY-015), High-Tech Research Key Laboratory of Zhenjiang City (No. SS2018002), and a Project Funded by the Priority Academic Program Development of Jiangsu Higher Education Institutions (PAPD), China.

**Institutional Review Board Statement:** Not applicable.

**Informed Consent Statement:** Not applicable.

**Data Availability Statement:** Data are contained within this article.

**Conflicts of Interest:** The authors declare no conflict of interest.

**Sample Availability:** Samples of the compounds are available from the authors.

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
