# Peer review of "Rationally Designed Ternary Deep Eutectic Solvent Enabling Higher Performance for Non-Aqueous Redox Flow Batteries"

_processes, doi:10.3390/pr10040649_

Round 1
Reviewer 1 Report
It was my pleasure to review the manuscript entitled: “” (manuscript ID: ). Text is constructed and edited correctly. However, I have some remarks:
1) Introduction should mention and compare (in terms of energy and power) RFB with other energy storage devices like supercapacitors (10.3390/polym11101648) and lithium-ion batteries (10.1002/adma.201800561)
2) There are some spelling mistakes like: “fourier” it should be written with capital letter, “polarity adjustion range” so I would recommend minor language check.
3) You should modify figure 6 b (Power density curve of ferrovanadium 228 flow battery in the two different ratios of DESs). It is a bit confusing and it is hard to decipher which points and lines are for which Y axis.
4) There are two figures 6! (on page 7 and 8)
5) What is missing in my opinion is some comparison to the current state-of-the-art regarding the similar RFBs
6) Figure 6 on page 8: there is a noticeable change in performance of RFBs between 1st and 5th cycle. What is the long-time preference (more than 5 cycle) of this RFB.
7) Again it would be grate to compare this RFB with other s in term of energy and power in form of a table or Ragone plot.
Based on my judgment I would recommend this work for a major revision.
Reviewer 2 Report
The reviewer had to read sentences repeatedly in order to assume or guess the meaning. Finally he stopped reading, no potential reader will be patient enough to guess the intentions of authors for several pages.

Reviewer 3 Report
All of my previous concerns have been addressed. The manuscript is acceptable for publication, subject to standard copyediting, with two minor corrections:
- Figure 1b, the y-axis should be "weight %" not "weight loss"
- Page 4, second paragraph, lines 134 and135, the term "weightlessness" is incorrect since the sample does have mass. I think what the author is referring to are areas of constant weight (or areas displaying no weight loss).
Reviewer 4 Report
The authors describe the identification and the use of a ternary DES in non-aqueous redox flow batteries.
The paper is relatively well written and illustrated in an adequate format for Processes with a subject that could be appropriated with the journal scope. However, the manuscript has to be carefully revised to avoid typing or English errors (for example lines 73 and 74 and throughout the entire manuscript).
The introduction is devoted to the description of batteries and their necessary characteristics for a good behaviour. Some improvements could be realized concerning especially the choice of the electrolytes (anolyte or catholyte).
Next, a (too) short description of DES and their application domains are given. Concerning the domain of biomass (line 69), the authors have to specify what is dealing with the biomass. TDES are then mentioned as well as redox pairs and their respective role. This part is interesting and appropriate. The viscosity is also introduced and discussed as a crucial factor for the study.
Concerning the part dedicated to the results, the voltammetry studies are nice and furnish important data. The graph involving both viscosity and conductivity in relationship with the molar ratio is also fine.
Concerning now the structures defined on figure 3, the use of ChemBio 3D Ultra is not suitable. Could the authors use better molecular modelling technics?
Could the authors also explain how they measured the diffusion coefficients (Table 2)?
The results obtained through FT-IR have also to be better explained and the conclusions too.
In conclusion, in this form, I recommend major revision before considering this MS for a publication in Processes.
Round 2
Reviewer 1 Report
I belive that the article can be accepted in the present form.
Author Response
Dear Reviewer:
Thank you very much for your comments and affirmation of our paper.
Reviewer 2 Report
The manuscript has been changed but not improved. Already the abstract turned out to be hardly comprehensible. Given the reviewers background he certainly can guess the meaning and intention of the authors, and he would happily help in making the maunscript suitable for publication. Not by adding one more time-consuming review.
Author Response

(The authors gave the same response as above.)

Reviewer 4 Report
The reviewer thank the authors for their work and this improved version of the manuscript.